# Overview on Therapeutic Options in Uncommon EGFR Mutant Non-Small Cell Lung Cancer (NSCLC): New Lights for an Unmet Medical Need

**DOI:** 10.3390/ijms24108878

**Published:** 2023-05-17

**Authors:** Giulia Pretelli, Calogera Claudia Spagnolo, Giuliana Ciappina, Mariacarmela Santarpia, Giulia Pasello

**Affiliations:** 1Department of Surgery, Oncology and Gastroenterology, University of Padova, 35128 Padova, Italy; giulia.pretelli@gmail.com; 2Medical Oncology Unit, Department of Human Pathology “G. Barresi”, University of Messina, 98122 Messina, Italy; spagnoloclaudia92@gmail.com (C.C.S.); giuliana.ciappina@gmail.com (G.C.); mariacarmela.santarpia@unime.it (M.S.); 3Oncologia Medica 2, Istituto Oncologico Veneto, IRCCS, 35128 Padova, Italy

**Keywords:** non-small cell lung cancer (NSCLC), epidermal growth factor receptor (EGFR), tyrosine kinase inhibitors (TKIs), uncommon mutation, compound mutation, intracranial activity, immunotherapy

## Abstract

The majority of epidermal growth factor receptor (EGFR) mutations (85–90%) are exon 19 deletions and L858R point mutations of exon 21, characterized by high sensitivity to EGFR-tyrosine kinase inhibitors (TKIs). Less is known about uncommon mutations (10–15% of EGFR mutations). The predominant mutation types in this category include exon 18 point mutations, exon 21 L861X, exon 20 insertions, and exon 20 S768I. This group shows a heterogeneous prevalence, partly due to different testing methods and to the presence of compound mutations, which in some cases can lead to shorter overall survival and different sensitivity to different TKIs compared to simple mutations. Additionally, EGFR-TKI sensitivity may also vary depending on the specific mutation and the tertiary structure of the protein. The best strategy remains uncertain, and the data of EGFR-TKIs efficacy are based on few prospective and some retrospective series. Newer investigational agents are still under study, and there are no other approved specific treatments targeting uncommon EGFR mutations. Defining the best treatment option for this patient population remains an unmet medical need. The objective of this review is to evaluate existing data on the outcomes, epidemiology, and clinical characteristics of lung cancer patients with rare EGFR mutations, with a focus on intracranial activity and response to immunotherapy.

## 1. Introduction

Despite the emerging treatment strategies of recent years, lung cancer remains the leading cause of cancer death worldwide, with an estimated 1.8 million deaths every year. Lung cancer is the primary cause of cancer-related deaths in men, and also the most commonly diagnosed cancer in this gender. On the other hand, among women, it ranks third in terms of incidence, after breast and colorectal cancer, but is the second leading cause of cancer-related deaths, with breast cancer being the top cause. According to the Human Development Index (HDI), lung cancer is the most commonly diagnosed cancer among men in countries with a higher HDI, with 39 cases per 100,000 people, and the second most common in countries with a lower HDI (10.3 per 100,000). Among women, it is the third most commonly diagnosed cancer in countries with a higher HDI (18.2 per 100,000), and the fifth most common in countries with a lower HDI (4.2 cases per 100,000). In terms of mortality, lung cancer is the leading cause of cancer-related deaths in men in both higher and lower HDI countries. Among women, it is the top cause of cancer-related deaths in countries with a higher HDI, and the fourth leading cause in countries with a lower HDI [1].

The majority of advances in the treatment of lung cancer have been made in the area of targeted therapies, with a particular focus on non-small cell lung cancer (NSCLC) patients with EGFR mutations. The use of targeted therapies has resulted in significant improvements in patient outcomes, leading to an improvement in both progression-free survival (PFS) and overall survival (OS) rates [2].

EGFR is a transmembrane receptor which has a crucial function in cancer cell proliferation, neoangiogenesis and the inhibition of apoptosis [3]. EGFR overexpression correlates with aggressive disease and poor prognosis [4], making it an optimal target for cancer therapy. The prevalence of EGFR mutations ranges from 14% in European patients to 38% in Chinese patients. EGFR mutation occurs mainly in adenocarcinoma histology, the female gender, and non-smoker patients [5]. The majority of EGFR mutations (85–90% of all EGFR-mutant patients) are deletion of exon 19 (ex19del) in the Leu Arg Glu Ala (LREA) residues (amino acid residues 747 to 750; 45% of EGFR mutations) and L858R point mutation of exon 21 (40%). These are known as activating EGFR mutations and are characterized by high sensitivity to EGFR-TKIs [6]. Exon 20 T790M is an uncommon mutation at NSCLC diagnosis, and it is mostly associated with about half of cases of resistance after first- or second-generation EGFR-TKIs [7,8].

Less is known about other mutations, which are defined as uncommon mutations, accounting for approximately 10–15% of all EGFR mutations (ranging between 1.0% and 18.2% across different series). These mutations usually show lower sensitivity to EGFR-TKIs, with some exceptions observed with the use of afatinib [9,10,11,12,13].

Unfortunately, uncommon mutations are not represented or underrepresented in most phase III clinical trials comparing EGFR-TKIs with chemotherapy or different EGFR-TKI generations. Available evidence on the drug sensitivity and treatment outcomes of EGFR-TKIs in patients harboring uncommon mutations is mostly retrospective and based studies of Asian patients, with a focus on the most frequent uncommon mutations (G719X exon 18; L861X exon 21; S768I exon 20) [11,14,15,16,17,18].

Although most clinicians agree and expert consensus recommends a front-line treatment with EGFR-TKIs instead of chemotherapy, defining and tailoring the optimal treatment strategy for this patient population is still an unmet medical need.

The aim of this review is to describe the epidemiology and clinical features of lung cancer patients affected by uncommon EGFR mutations, and discuss available data on the outcome of patients receiving different treatment options.

## 2. Epidemiology of Uncommon EFGR Mutation

As mentioned above, the most frequent types of EGFR mutations are ex19del in the LREA motif and mutation L858R, characterized by high sensitivity to EGFR-TKIs [6], and mutation T790M, which is associated with 40–55% of cases of resistance to first- or second-generation EGFR-TKIs [7,8]. Significantly less evidence is available on mutations other than ex19del, L858R and exon 20 T790M, defined as uncommon mutations and accounting for about 10–15% of all EGFR mutations. John et al. analyzed the prevalence of uncommon EGFR mutations across ten studies, mostly conducted in China, showing an occurrence rate of uncommon mutations ranging from 1.0% to 18.2% [10].

The clinical features of this patient population are similar to patients with common EGFR mutations, although some studies have shown an association with smoking history and older age [9,19].

In a series of 5363 Chinese patients, the frequency of EGFR mutations was found to be 34%. Among these patients, the frequency of uncommon mutations was 11.9%. It was observed that there were more male patients (54.1% vs. 44.4%) and smoker patients (30.7% vs. 24.3%) in the group with uncommon mutations compared to the group with common mutations. Statistically significant differences were observed, with *p*-values of 0.007 and 0.039, respectively [18]. Some mutation types predominate within this group, such as point mutations in exon 18 at position G719X (0.9–4.8%), the exon 21 L861X mutation (0.5–3.5%), insertions in exon 20 (ex20ins; 0.8–4.2%), and S768I in exon 20 (0.5–2.5%) [10]. Other uncommon alterations comprise exon 18 indel (i.e., pE709_T710delinsD), mutations involving codon 709 of exon 18 (such as pE709K/A/G/V), as single or complex mutations, EGFR amplification, exon 21 missense, exon 19 insertions, and EGFR variant III [20,21]. Although less frequent than activating mutations, the prevalence of some uncommon mutations is comparable to that of other druggable alterations, such as RET or ROS1 fusions [19].

Furthermore, the prevalence of EGFR mutations in exons 18–21 varies geographically. Graham et al. conducted a survey on EGFR testing performed in selected laboratories worldwide over the course of a year. The survey included 170 clinical laboratories from 20 different countries, accounting for a total of 136,533 tests. The survey found that mutation prevalence was 30–46% in Asia, 16% in Africa and in the Middle East, 13% in Europe, and 8–9% in North and South America. The L861Q mutation and exon 20 mutations were less frequently detected, as expected. Exon 20 mutations were more prevalent in Africa and the Middle East, while L861Q was more prevalent in Northern Asia. The low number of samples for these uncommon mutations precluded statistical analysis [22]. The prevalence of rare EGFR mutations is made even more heterogeneous by the different testing methods and type of reports, referring to uncommon mutations as single drivers or within compound mutations [23].

While some reports show that G719X, L861Q and S768I confer sensitivity to EGFR-TKIs, albeit with a lower treatment response than common mutations, ex20ins are known to be resistant to first- and third-generation EGFR-TKIs [6,10,20,24].

There are no approved TKIs or established guidelines for the treatment of this subgroup of patients, and the standard of care is chemotherapy [24].

## 3. EGFR Mutation Testing Methods

The incidence of uncommon EGFR mutations has increased during the last decade, and this is likely due to the improvement of detection methods, with particular reference to the use of next-generation sequencing (NGS) [19].

Over the years, an increasing number of methods have become available to determine EGFR gene mutations. Some of them can identify the most common genetic alterations, but miss out on other mutations, emphasizing the importance of sequencing-based techniques to detect uncommon mutations [25,26]. Available evidence has shown that Sanger sequencing and real time PCR (RT-PCR) have a lower detection rate for single, uncommon or compound EGFR mutations compared to NGS [26,27].

Sanger sequencing is used for the detection of single nucleotide variants, insertions, and deletions in clinical practice, and it still represents the gold standard for such uses [23]. However, it has some limitations due to low sensitivity (15–20%) [23,28,29]. Mao et al. have shown that the detection rate of Sanger sequencing is significantly lower when compared to NGS and RT-PCR [30]. The RT-PCR technique is based on the use of fluorescent probes to specifically amplify known mutations. However, this method has some limitations, since it may not detect uncommon or compound mutations [23,25]. The other techniques used are the pyrosequencing system and digital PCR [20,23]. The pyrosequencing system is a quantitative methodology based on the clonal amplification of emulsion PCR and the subsequent detection of light signals from the DNA growing chain [31]. Digital PCR is a technique able to detect and quantify a target molecule in a precise and accurate manner [32]; in particular, digital droplet PCR (ddPCR) has shown a high sensitivity. In a previous analysis of Gu et al., this sensitivity was 96% for ddPCR, in comparison to RT-PCR and NGS [33].

Finally, NGS is able to detect an increasing number of EGFR mutations and concomitant alterations. Several NGS panels are currently available, allowing for concurrent evaluation of several target hotspots [23]. The NGS technique on cell-free DNA (cfDNA) may eliminate the need for biopsies (which can be difficult to obtain in some cases). A previous study exploring NGS analysis on ctDNA showed a sensitivity of 75% and a specificity of 100% [34].

NGS showed benefits in comparison to Sanger sequencing and RT-PCR by providing the highest number of EGFR mutations and by identifying other non-EGFR mutations with potential targeted drugs [30].

NGS allows us to detect not only classical EGFR mutations, but also a broad number of concomitant mutations, rare mutations and mechanisms of resistance that can impact treatment outcomes and improve therapeutic options for patients [23]. However, no large prospective trials have yet evaluated the clinical impact of detecting rare and compound mutations, and further evidence is needed [20,21].

As per ESMO (European Society for Medical Oncology) guidelines, it is currently recommended to use the NGS technique for patients with advanced non-squamous NSCLC (plasma or tumor sample) in order to detect ESCAT (ESMO Scale for Clinical Actionability of molecular) level I alterations, for which a drug validated in clinical trials is available, thereby driving treatment decision [35].

This may not always be feasible in a real world scenario. In particular, when considering the heterogeneity of mutations such as ex20ins and limitations in clinical testing methods. Baumi et al. studied a sample of 175 patients with ex20ins detected by NGS, and noticed that only 89 (50.9%) would have been identified by a PCR test. A second dataset of 627 patients provided confirmation of this concern; as per the results, PCR testing was expected to overlook 51.4% of ex20ins cases that were detected by NGS [36].

## 4. From Exon-Based to Structure-Based Classification of EGFR Mutations

The EGFR gene can harbor different mutations, and although several of them may appear similar to classical ones, their response to EGFR-TKIs, as well as any resistance to treatments, may be heterogeneous and difficult to predict [23].

In this context, a predictive system for classifying EGFR mutations based on their sensitivity would be of crucial interest in order to guide treatment decisions [23,24]. Robichaux et al. studied a large database of EGFR mutant NSCLC, developing a new preclinical model of 76 different EGFR mutations treated with 18 different EGFR inhibitors (first-, second- and third-generation, as well as ex20ins TKIs) [24,37]. Based on their response to EGFR-TKIs, a new structured-based classification has been outlined, stratifying non-classical EGFR mutations into four main subgroups: classical-like mutations (distant to the ATP-binding pocket), T790M-like mutations (within the hydrophobic core), insertions in the C-terminal end of αC-helix in exon 20, and finally, mutations within the ATP-binding pocket or C-terminal end of the αC-helix, which compress the P-loop and the αC-helix itself (PACC mutations). A simplification of the tertiary structure of EGFR is shown in Figure 1 [24].

This structure–function-based classification seems to better identify drugs’ sensitivity compared with a simple exon-based classification; thus, it is possible that mutations at different gene sites may induce similar changes in the tertiary structure of the receptor, such as in case of PACC mutations [24,37].

It has been observed that classical-like, atypical EGFR mutations have a small effect on the EGFR global structure when compared with wild-type EGFR, and are sensitive to all EGFR-TKIs [24].

Robichaux et al. found that not all exon 20 mutations exhibit the same response to EGFR-TKIs. Exon 20 point mutations, which belong to the PACC mutation subgroup, were sensitive to second-generation EGFR-TKIs, in contrast to the majority of ex20ins located in the αC-helix, which behave similarly to “classical-like” mutations. In contrast, ex20ins in the C-terminal loop of the αC-helix appeared to be more sensitive to second-generation EGFR-TKIs. In particular, the mutations nearer to the C-terminal loop were found to be more sensitive to EGFR-TKIs than the farther ones [37]. In addition, it was found that in the case of a classical EGFR mutations co-occurring with a PACC mutation, the model seemed to predict a response to the second-generation EGFR-TKIs [24,37]. Such evidence needs to be further confirmed, hopefully in prospective clinical trials, as it could have important clinical–therapeutic implications for EGFR mutant NSCLC [37].

## 5. Compound Mutations

Heterogeneous outcomes in patients harboring uncommon mutations may also result from the co-occurrence of uncommon mutations within compound mutations both associated with common and uncommon alterations [38,39]. Thus, it can be assumed that the presence of co-occurring alterations contributes to the increased spatial and temporal heterogeneity of EGFR-mutant NSCLC [40,41], as some subclones may gain a proliferative advantage under treatment pressure, leading to acquired resistance that may arise sooner or later. In a recent study, it was found that EGFR compound mutations were virtually homogeneous both inter- and intratumor in a small series of patients. As a result, the optimal treatment should be chosen based on the specific EGFR mutation detected, including the type of compound mutations [39]. Attili et al. proposed four main categories of compound EGFR mutations: combined common EGFR mutations (exon 21 p.L858R + exon 19 deletions), combined common (exon 21 p.L858R + exon 19 deletions) plus uncommon EGFR mutations (any but exon 21 p.L858R, exon 19 deletions or de novo exon 20 p.T790M), combined uncommon EGFR mutations and combined EGFR mutations (any), plus de novo exon 20 p.T790M (Table 1) [42].

Compound EGFR mutations are represented as double or multiple nonsynonymous mutations of the EGFR tyrosine kinase domain, where a typical EGFR mutation (i.e., ex19del, L858R) in the majority of the cases is identified together with an uncommon mutation, or a combination of two uncommon mutations [6,38]. The clinical significance of compound mutations is still unclear, and they are frequently detected with advances in sequencing technology, including NGS [38].

The incidence of compound mutations is highly heterogenous, and varies across the studies from 3% to 26% of total EGFR mutant cases; this heterogeneity is probably dependent on the different testing methods used, the patient population, and the specific mutations considered [43,44].

A study conducted by Kim et al. found compound EGFR mutations in 24.6% of the cases of EGFR-mutant lung adenocarcinoma, and the majority of them were represented by a combination of the atypical mutation and typical mutation. Examples of partner alterations were mutations in exon 18 (V689L, I706T, and E709K), in exon 20 (H773Y and R776H), and in exon 21 (L833V, H870R, and A871G). One patient harbored a compound mutation of L858R and ex19del. Kim et al. also found that patients with compound mutations were most likely to have a higher burden of missense mutations. It was observed that the patients with compound mutations had shorter OS than those with simple mutations (83.7 vs. 72.8 months). Thus, there is a need to closely monitor these patients during follow-up. The subtypes associated with poor clinical outcomes, such as papillary/micropapillary types and the solid with mucin production type, were more frequently detected in cases with compound mutations. OS was significantly poorer in the cases with compound mutations, but there was no difference in the duration of disease control between groups with compound or simple mutations treated with EGFR-TKIs upon the recurrence [38].

Kobayashi et al. found compound EGFR mutations in 14% of the patients included in their study; most patients had an EGFR sensitizing mutation (i.e., G719X, ex19del, L858R and L861Q) and an atypical mutation. Reporting the genotype–response pattern of NSCLC with EGFR compounds and uncommon mutations will be helpful in guiding appropriate decision-making for the treatment of patients with EGFR-mutant NSCLC [6].

Preclinical data suggest that patients with NSCLC harboring EGFR compound mutations are associated with different sensitivity responses to different TKIs [39]. A few papers have reported the presence of different responses to the EGFR-TKIs among patients harboring compound EGFR mutations [38].

In a series of 106 patients receiving first-generation EGFR-TKIs, the median progression-free survival (mPFS) of patients with compound mutations was significantly poorer compared to patients with a single common mutation (9.1 vs. 13.0 months, *p* < 0.001). Furthermore the response rate (RR) to the treatment of patients harboring compound mutations was lower than that of patients with single common mutation, though without statistically significant difference (50.9% vs. 67.8%, *p* = 0.088). Within the group harboring compound mutations, the patients with double rare mutations (i.e., co-occurring mutation in exon 20) had worse mPFS than patients with other compound mutations or a common mutations (6.5 vs. 9.1 vs. 13.0 months, *p* = 0.002) [43]. Other evidence suggests that patients with two common EGFR mutations treated with first-generation EGFR-TKIs had a similar response rate and PFS to patients with a single common mutation [44,45].

The objective response rate (ORR) and PFS to TKIs vary significantly In patients with either common and rare mutations, and while those with single exon 20 mutations are typically resistant to TKIs [43], it remains uncertain whether patients with an EGFR exon 20 mutation accompanied by another mutation are candidates for TKI therapy; previous reports have shown that patients with ex20 compound mutations had a response to EGFR-TKIs, while other patients with single ex20 mutation had progressive disease (PD) upon first evaluation [46,47].

Besides compound mutations within the EGFR gene, different co-mutations are mostly present with very rare EGFR mutations, and the more frequent TP53 seems to have a detrimental effect on TKIs’ treatment outcomes. This certainly emphasizes the importance of understanding tumor heterogeneity for determining the treatment sequence [48]. Another mutation in a different gene is PIK3CA, which drives resistance to EGFR-TKIs by activating bypass AKT signaling; it is found in 4% of patients with lung cancers [41] and in 3.5% of EGFR mutant NSCLC [49].

## 6. Treatment Activity Data of Different TKIs

### 6.1. The More Common among Uncommon: L861Q, G719X and S768I

Three generations of EGFR-TKIs have been introduced in the clinical practice as the standard of care for common EGFR mutations [50,51]. These molecules have different pharmacological characteristics and mechanisms of action; the first-generation TKIs, erlotinib and gefitinib, are reversible EGFR inhibitors [52,53,54,55,56,57]. They prevent EGFR from undergoing auto-phosphorylation, which in turn stops downstream signaling by competitively engaging with the ATP-binding region. The second-generation TKIs, afatinib and dacomitinib, bind to the EGFR kinase domain via covalent, irreversible bonds, and may be more active against other receptors of the ErbB receptor family [55,56,58,59]. The third-generation irreversible TKI, osimertinib, has been designed to specifically target the gatekeeper T790M mutation, which confers resistance to first- and second-generation TKIs by interfering with the bond to the ATP-binding site [60]. The optimal treatment of patients with tumors harboring uncommon EGFR-activating mutations remains uncertain. 

Data about the efficacy of EGFR-TKIs in patients with NSCLC harboring uncommon EGFR mutations are limited to a few prospective studies with afatinib (LUX-lung 2, 3 and 6) [14], one prospective study with osimertinib (KCSG-LU15–09) [61], and several retrospective series and case reports [15,16,17,18].

The available data show clinical activity and efficacy in treating the mutations G719X, L861Q, and S768I [14], for which first-line treatment with EGFR-TKIs (particularly afatinib) has been shown to significantly improve the PFS compared to first-line chemotherapy [62,63].

The exon 20 point mutation pS768I showed a good response to afatinib (a median PFS of 14.7 months) in the trials LUX-lung 2, 3 and 6 [14], and a PFS of 12.3 months in patients treated with osimertinib in a recent trial [61]. One real-world study with afatinib focusing on Chinese patients showed a prevalence of 12% of uncommon mutations; the entire patient population harboring uncommon mutations had a PFS of 9.06 months [64]. A recent large study on a database of 693 EGFR mutant patients harboring 98 different uncommon mutations explored the efficacy of afatinib; the data have been collected from randomized clinical trials and phase IIIb trials, compassionate-use/expanded-access programs, noninterventional trials, case series or case studies [65,66]. For the 272 untreated patients harboring the mutations G719X, L861Q, and S768I, the median time to treatment failure (TTF) was almost 1 year. In contrast, for patients with ex20ins and other uncommon mutations, the median TTF was 4.2 and 4.5 months, respectively. Afatinib showed efficacy even in patients with compound mutations, with a median TTF of 14.7 months, and this was even higher if one of the mutation was common (16.6 months) [65]. Another study confirmed the clinical activity of afatinib for patients with compound EGFR mutations, and a better PFS compared to gefitinib and erlotinib. In 2018, the Food and Drug Administration (FDA) approved afatinib for the treatment of patients harboring the following uncommon EGFR mutations: L861Q, G719X, and S768I, based on a combined analysis of the aforementioned LUX-lung 2, 3 and 6 trials [14,67].

Preclinical data from NSCLC models harboring these three uncommon mutations suggest a clinical activity of osimertinib in this setting [68]. Results from the prospective phase II study KCSG-LU15-09, with first line osimertinib in patients with NSCLC with uncommon EGFR mutations, showed an ORR of 50%, a median PFS of 8.2 months, and a median OS not reached [61]. Few data are available regarding the outcomes of osimertinib in patients with uncommon EGFR mutations in the real world [66]. A retrospective study showed the activity of osimertinib in patients with NSCLC harboring uncommon mutation, although with less clinical benefit compared to those with common mutations. L861Q and ex19delins have better outcomes [69].

### 6.2. Focus on Exon 18

Exon 18 mutations account for 3–4% of EGFR mutations, and comprise mutations in codon 719 (G719A/S/C) and 709 (E709X), and less frequently, del-ins [19].

Mutation G719X, after ex20 ins, is the most frequent uncommon mutation, and although heterogeneously, it shows sensitivity to TKIs, in particular a high ORRs (75–78%) with afatinib [14] and neratinib [70]; with respect to first-generation TKIs, this is comparable to the response of common mutations [19]. Mutations involving codon 709, such as pE709K/A/G/V, as single or complex mutations, are known to be resistant to first-generation TKIs; however, some of them are sensitive to afatinib (pE709K/A) [20,71,72], and generally occur as part of a compound mutation [15].

The most common ex18 deletion is delE790_T710insD, and in a preclinical model it has been shown to be the least sensitive to EGFR-TKIs among the ex18 mutations [73]; very few clinical data are available, showing some activity of afatinib [15]. Patients may harbor del-ins with other uncommon mutations (ex20 T790M) [19,74].

### 6.3. Focus on Exon 19

Mutations in ex19 are the most common EGFR mutations, but their sensitivity to EGFR-TKIs varies largely; the deletions in LRE fragment (L747 to E749) are known to be more sensitive to EGFR-TKIs, while non-LRE deletions have a lower response to EGFR-TKIs [75]. Uncommon ex19 deletion–insertion variants (ex19delins) account for 5% of EGFR mutant NSCLC, and have different sensitivity to EGFR-TKIs [76]. Some variants have a similar structure to that of ex19del, have reported sensitivity to first/second-generation EGFR-TKIs in vitro and in vivo [77], and have significantly better PFS when treated with first-generation TKIs compared to the common ex19del. The most common variant is L747_A750delinsP, known to be sensitive to afatinib [76], and similar to some ex19del variants between aminoacid residues 745–753, showing sensitivity to TKIs [20]. Other exon 19 insertions such as p.L747S, p.D761Y and p.T854A confer resistance to EGFR-TKIs [78]. Interestingly, patients with uncommon ex19delins showed a better PFS than patients with common ex19del; nevertheless, when treated with first-line EGFR-TKIs, the two groups had a similar risk of developing resistance by acquiring the T790M mutation. Subsequently, when treated with osimertinib second-line, the patients with ex19delins showed a significantly poorer outcome (except in the case of variant L747_A750P) [76].

### 6.4. Focus on Exon 20

Ex20ins is the largest group among uncommon EGFR mutations, consisting of insertions or duplications within 15 amino acid residues (761–775), with heterogeneous responses to EGFR-TKIs, the vast majority being resistant [20,79]. The residues 761–766 code for the C-helix of the protein, while residues 767–775 code for the loop following the C-helix [78]. The differences in the structure are supposed to be the cause of the heterogeneous response to EGFR-TKIs [19]. Indeed, some data have shown a promising response to afatinib [65]. Preclinical evidence has shown that insertions in codons 769 to 775 could lead to drug resistance, while those in more proximal codons might have a similar structure to classical mutations [20]. One of the most frequent mutations (5–6% of ex20ins) is p.A763_Y764insFQEA, which confers to the protein a structure similar to that of the L858R mutation, and shows a response to erlotinib (partial response or stable disease) [47,80]. Another ex20 mutation, p.A767_V769dupASV, which is identical to p.V769_D770insASV, showed some preclinical activity, in terms of tumor growth inhibition, in response to afatinib combined with cetuximab; however, clinical evidence is lacking [81]. Different types of ex20ins were found to be sensitive to afatinib: p.773_774HVinsGHPH, p.A767delinsASVD39, and p.A767_S768insSVA [62,65]. On the contrary, the mutation p.D770_N771insSVD confers low sensitivity to all TKIs [20], thus confirming the heterogeneity of patients harboring ex20ins [6]. The acquired point mutation in exon 20 p.C797S, together with T790M, is the most common mechanism of resistance to third-generation TKIs. When the mutation is detected in trans, a combination of first- and third-generation TKIs could result in clinical efficacy. When it is detected in cis, it confers resistance to TKIs in combination or alone [20], suggesting a significant impact on the tertiary structure of the protein. Before the introduction of novel drugs targeting ex20ins, the gold standard of treatment for this subgroup of patients was platinum-based chemotherapy [19]. However, in recent years, novel treatment strategies have become available for patients with ex20ins.

Poziotinib is a novel EGFR-TKI that has been studied in a phase II trial, which showed clinical activity in patients with EGFR ex20ins and HER2 ex20ins [82,83]. The small size of the drug and its flexibility are the key to its effectiveness against these mutations, limiting the TKI bonding site [83]. Although effective, the expanded access program showed a high rate of toxicity, with 66% of patients reporting grade 3 adverse events (AEs) and dose interruptions, which has limited its current clinical development [84]. Another TKI specifically targeting ex20ins is mobocertinib, a selective EGFR/HER2-TKI, oral and irreversible; it demonstrated significant benefit in NSCLC patients pretreated with EGFR ex20ins [85]. Mobocertinib received accelerated approval from the FDA in September 2021 [86]. Finally, amivantamab is a bispecific monoclonal antibody targeting MET and EGFR, which was approved by the FDA in May 2021 and by the European Medicines Agency (EMA) in December 2021 [87,88], for the treatment of patients with NSCLC harboring ex20ins. This approval was based on the results of the CHRYSALIS trial, which demonstrated durable efficacy and a manageable safety profile. The ORR achieved by the study population was 40%, with a mPFS of 8.3 months. The majority of adverse events observed in the study were rash (86%), followed by infusion-related reactions (66%) and paronychia (45%). Some 5% of patients developed G3–4 hypokalemia, 13% of the patients required a dose reduction, and 4% discontinued the treatment [89]. Focusing on infusion-related reactions (IRR), Park et al. noticed that these were a frequent AE, but mostly G1–2, limited to the first administration and treated with antihistamines, steroids, antipyretics and infusion holding. Subsequent infusions were not affected by the initial IRR, and only 1% of patients discontinued their treatment due to this AE [90]. Newer investigational agents that are still under study in clinical trials have demonstrated promising results in the treatment of patients with ex20ins. Sunvozertinib (DZD9008) is a novel, irreversible EGFR and HER2 TKI under investigation in phase I/II studies (NCT03974022, CTR201920) that has showed antitumor activity in different types of ex20ins, with a ORR of 39.3% [91]. Other novel drugs under study that have revealed clinical activity in patients harboring EGFR ex20ins include CLN-081 (TAS6417) [92] and tarloxotinib [93].

The toxicity and response rates of the inhibitors listed above are detailed in Table 2.

According to ESMO consensus, the first-line treatment for this patient population should be platinum-based chemotherapy, followed by amivantamab or mobocertinib as second-line treatment. The use of immune checkpoint inhibitors (ICIs) is not a priority due to the risk of toxicity and uncertain evidence [78].

### 6.5. Focus on Exon 21

The most frequent uncommon mutation in exon 21 is p.L861Q (1–2% of all EGFR mutations), which has been demonstrated to be sensitive to afatinib and osimertinib [14,61]. Other less frequent mutations, such as p.A864T and p.L861R, appear to be sensitive to afatinib and osimertinib with in vitro models [20].

Other rarer mutations with generally low sensitivity are L862V, V851X, A859X, while the response is uncertain for E866K, H825L, P848L, H870Y/R, and G836S [47,94,95,96].

## 7. Intracranial Activity of Different EGFR-TKIs in Uncommon Mutations

The brain represents one of the most common sites of metastases for patients with NSCLC, occurring in about 40% of EGFR mutant cases during the course of the disease, and thus representing a clinical challenge. Most EGFR-TKIs show a lack of evidence regarding intracranial activity [97]. The FLAURA trial demonstrated that osimertinib has a potent activity against brain metastases (BMs) in patients with EGFR-mutated NSCLC compared to erlotinib or gefitinib [98]. It also showed promising efficacy in patients with the de novo T790M mutation. The clinical outcome of patients harboring uncommon EGFR mutations and BMs and treated first-line with TKIs remains unclear [97]. Evidence suggests that patients with uncommon EGFR mutations have a significantly higher prevalence of BMs [99], and first-line EGFR-TKIs appear to be less effective in controlling and preventing BMs in this patient population [97].

A previous study showed benefits from treatment with afatinib, but with limitations due to the small number of the patients treated. Seven patients with uncommon EGFR mutations and BMs (leptomeningeal metastases or brain parenchymal metastases) were treated with TKIs, of whom four responded positively to the treatment as detected by MRI and CEA levels. Three patients received afatinib, and one patient received osimertinib [100].

A retrospective study that examined EGFR-mutant NSCLC patients receiving first-line EGFR-TKIs (86% gefitinib and erlotinib, 4.8% afatinib and 9.2% osimertinib) showed that among the group of patients with baseline brain metastases, those harboring uncommon mutations had a significantly shorter intracranial time to progression compared to patients with L858R mutation (23.6 months vs. 68.0 months, *p* = 0.003) and ex19del (23.6 months vs. NR, *p* < 0.001). Furthermore, patients with uncommon EGFR mutations had a higher risk of intracranial PD, suggesting the need to implement treatment strategies in order to prevent and control BMs [97].

Furthermore, emerging data suggest significant intracranial activity with second-generation TKIs, particularly dacomitinib. Among the 32 patients included in the study conducted by Zhang et al., 30 were evaluable with measurable or non-measurable central nervous system (CNS) lesions; the intracranial ORR was 66.7% (95% CI 47.2–82.7%), and the intracranial disease control rate (DCR) was 100% (95% CI 48.7–95.7%). Median intracranial DOR and median intracranial PFS were not reached. The study showed a significant CNS efficacy of dacomitinib in patients with EGFR-mutated NSCLC treated first-line in the real-world setting [101].

Another recent study exploring the activity of dacomitinib in patients with EGFR mutant NSCLC with BMs included one patient with G719A and I706T co-mutations; the patient had a CNS response and an overall partial response (PR) to the treatment [102]. Finally, regarding the new drugs under study, it has been observed that the aforementioned CLN-081 may demonstrate intracranial activity. Three patients treated with brain metastases have been reported; one patient achieved a stable disease (SD), and one patient obtained a PR [103]. Considering the drugs approved for ex20ins, mobocertinib, despite being a small molecule, appears to have a low brain penetrance, as shown by the worse confirmed objective response rate and high number of brain PD (25%) presented by the patients with brain metastases, compared to the group without CNS disease, in the phase I/II trial [104].

Due to its large molecular size, amivantamab is unlikely to cross the blood–brain barrier, and is expected to have poor activity in treating brain metastases; therefore, its clinical use as a monotherapy may be limited in patients with brain metastases [105]. To address this challenge, studies with combination therapies are ongoing, such as CHRYSALIS-2, assessing amivantamab and lazertinib versus lazertinib monotherapy in patients with EGFR-mutant NSCLC, and will include patients with treated brain metastases [106].

## 8. Response to Immunotherapy and Chemoimmunotherapy

With some exceptions within the group of ex20ins, EGFR-TKIs seem to be the best treatment option for uncommon EGFR mutations, and recent data confirm a modest activity of ICIs [19,107]. This is probably due to the tumor immune microenvironment of EGFR mutant NSCLC, which is associated with uninflamed characteristics, low PD-L1 (programmed death-ligand-1) expression/CD8+ TILs (Tumor-infiltrating lymphocytes), and a low tumor mutational burden (TMB) [108,109]. Interestingly, some patients with a history of smoking and high PD-L1 expression [110], despite harboring an EGFR mutation, may benefit from treatment with ICIs, especially patients harboring uncommon mutations [66,74,110]. Recently, an association between high PD-L1 expression and uncommon EGFR mutation has been shown (Figure 2) [74].

Moreover, a recent study demonstrated that treatment with EGFR-TKIs may change the tumor microenvironment by increasing PD-L1 expression (Figure 3) and TMB, while modifying CD8+/FOXP3 TILs and CD73 expression. Patients who had high PD-L1 expression after TKI treatment achieved a longer PFS after subsequent treatment with ICIs (pembrolizumab or nivolumab), of 7.1 months vs. 1.7 months, respectively, with a *p*-value of 0.0033, which is statistically significant [111].

Focusing on ex20 mutations, a recent study has shown that in this subgroup of patients, a tumor immune infiltration is evident, suggesting a role for ICIs [112]. A retrospective study conducted on patients with EGFR mutant NSCLC who were treated with ICIs found that those with ex20ins had a better RR, DCR, and PFS than those with common EGFR mutations [113]. This could potentially be attributed to the fact that patients with uncommon mutations tend to have a higher TMB [107]. On the contrary, patients who acquire the T790M mutation have a poorer prognosis when treated with ICIs, as well as with a combination of ICIs and chemotherapy, compared to those with other acquired resistance mechanisms. This is likely because patients with acquired resistance to TKIs (other than T790M) may exhibit higher levels of PD-L1 [114].

The majority of clinical trials with ICIs, including oncogene-addicted NSCLC, did not report details about the types of EGFR mutations or uncommon mutations [19]. The immunotarget registry, which includes a considerable percentage of patients with uncommon or compound mutations receiving ICIs, showed a response rate of 12%, and a median PFS and OS of 2.1 and 10 months, respectively. Some series have shown higher PD-L1 expression and better survival in patients with uncommon compared to common mutations [115]. Additionally, the hypothesis that patients with uncommon mutation and without T790M mutation could have a better response to ICIs was supported by a retrospective analysis by Yamada et al., which showed a significant correlation between the mutations G719X and ex20ins and outcome [116].

The combination of chemotherapy and immunotherapy seems to be more promising compared with single agent ICI in patients with EGFR mutant NSCLC, although previous series included only a small proportion of uncommon mutations [114,117]. The Impower 150 trial, which explored atezolizumab combined with carboplatin/paclitaxel/bevacizumab, showed better OS compared to the same regimen without ICIs for patients with EGFR mutations, including uncommon ones [118,119]. These data suggest that this combination could represent a therapeutic option for EGFR-mutated NSCLC. However, recent results from the final analysis showed a loss of statistically significant OS [120]. One retrospective observational study that explored the efficacy of ICIs or ICIs in combination with chemotherapy in EGFR mutant NSCLC included a patient population of whom 13% had uncommon EGFR mutations, in particular ex20ins, G719X, one L861Q. The group of patients treated with chemotherapy plus ICIs had a longer PFS (4.23 vs. 2.93) and OS (not reached vs. 19.67) compared to ICIs, but the differences were not statistically significant, with *p*-values of 0.599 and 0.270, respectively (Figure 4) [114]. Recently, a potential role for pembrolizumab has been hypothesized in patients with the G719X mutation and high PD-L1 expression (≥50%). However, the small number of patients involved does not allow for conclusions to be drawn [74].

## 9. How to Define a Treatment Sequence

The treatment sequence should be defined by first considering the resistance mechanisms and the available treatments for the subpopulation of EGFR mutant lung cancer. Acquired resistance mechanisms to first/second and third-generation TKIs are different, and mainly subclassified in the following three categories: mutations in target genes (on-target mutations), alternative pathway activation (off-target mutations) and histological transformation [121]. Resistance mechanisms to first and second-generation TKIs after PD are more commonly EGFR-dependent (i.e., the T790M mutation, accounting for 50% of cases with acquired resistance to gefitinib or erlotinib, and second-point mutations such as D761Y, T854A, or L747S), and are more heterogeneous and EGFR-independent after osimertinib (due to MET/HER2 amplification and the activation of the MAPK or PI3K pathways) [60]. Moreover, uncommon mutations may emerge within compound mutations as resistance mechanisms that drive treatment decisions to switch to a different generation of TKIs. Finally, osimertinib seems to have the best safety profile compared to other TKIs, showing a lower incidence of grade 3 or 4 AEs compared to first or second-generation TKIs [122]. With regard to afatinib, a systematic review and meta-analysis showed that second-generation TKIs had a comparable rate of grade 3 or 4 AEs with respect to erlotinib, but a rate greater than gefitinib [123]. This is confirmed by the LUX-lung 7 trial, which showed a greater rate of AEs of grade 3–4; however, the overall incidence was comparable [124].

In order to define the best treatment sequence in this setting, we have to take into consideration the available survival data derived from the literature. Unfortunately, we do not have randomized trials comparing second and third-generation TKIs; however, data from the GioTag study have demonstrated that a sequence of second and third-generation TKIs can achieve a clinically significant survival, although no uncommon mutation was included [125].

A recent multicenter cohort study including a small subgroup of patients with uncommon EGFR mutations showed no significant difference in survival in the overall population receiving afatinib compared with osimertinib. However, a better outcome was observed with osimertinib, particularly in patients with brain metastases [99]. Nevertheless, emerging data, as already mentioned, suggest intracranial activity with second-generation TKIs, particularly dacomitinib [101,102].

## 10. Ongoing Clinical Trials

The solution to the best treatment sequencing for patients harboring uncommon EGFR mutations may come from the ongoing phase II study CAPLAND (NCT04811001), which is exploring the best treatment sequencing of dacomitinib followed by or subsequent to osimertinib in patients with NSCLC harboring classical or uncommon EGFR mutations; furthermore, the efficacy of dacomitinib will be defined in patients with BMs.

Among other EGFR-TKIs, lazertinib, a new third-generation EGFR-TKI, is currently under investigation in combination with amivantamab, a bispecific antibody targeting MET and EGFR, in a phase I/Ib CHRYSALIS-2 study in patients with EGFR mutant NSCLC in progression on osimertinib. The cohort C of the study includes patients with uncommon mutations other than exon 20 insertion [106].

The updated results presented at the 2022 American Society of Clinical Oncology (ASCO) Annual Meeting showed that the combination demonstrates durable activity after progression on both chemotherapy and osimertinib [126]; the combination demonstrated an ORR of 33%, with a median DOR of 9.6 months. The phase III trials MARIPOSA and MARIPOSA-2 are currently ongoing, evaluating amivantamab in combination with lazertinib as first-line treatment and in combination with carboplatin and pemetrexed after PD to osimertinib.

## 11. Conclusions

The optimal treatment strategy for NSCLC patients harboring uncommon EGFR mutations remains an unmet medical need. Ongoing clinical trials will attempt to define the most effective therapeutic sequence for various mutation subgroups in the near future. However, until the outcomes of these trials are available, the best treatment approach for these patients must consider multiple factors. The treatment plan should take into account the activity and efficacy data specific to rare mutations, which can differ from those of common mutations. Furthermore, when designing the optimal treatment strategy, it is important to consider the safety of the available drugs and the acquired resistance mechanisms. By carefully considering these factors, healthcare professionals can provide the best possible treatment for patients with uncommon mutations until the results of ongoing clinical trials become available.

## Figures and Tables

**Figure 1 ijms-24-08878-f001:**
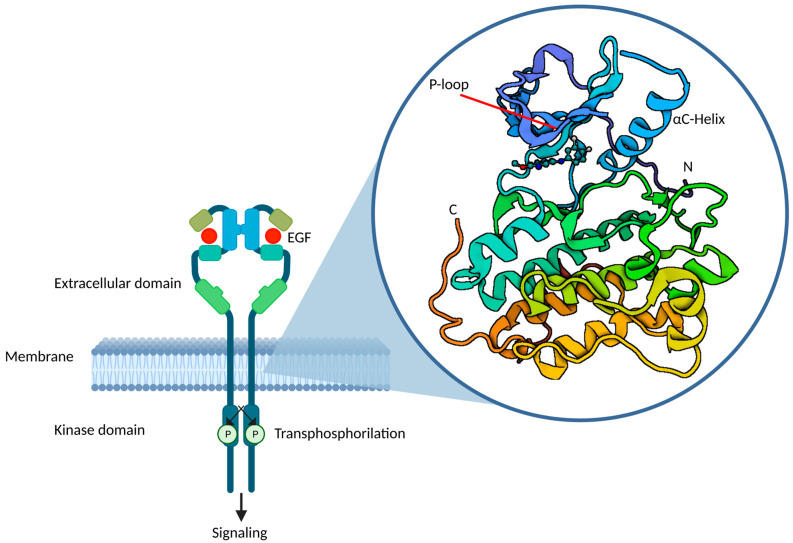
Simplification of the tertiary structure of EGFR; C-terminal and N-terminal, αC-helix, p-loop. EGF: epidermal growth factor.

**Figure 2 ijms-24-08878-f002:**
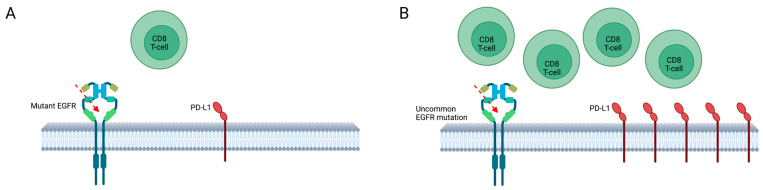
Tumor microenvironment of EGFR common (**A**) and uncommon (**B**) mutations. Association between high PD-L1 expression and uncommon EGFR mutations; larger PD-L1 overexpression in patients with uncommon compared to common EGFR-mutations. PD-L1: Programmed death-ligand-1.

**Figure 3 ijms-24-08878-f003:**
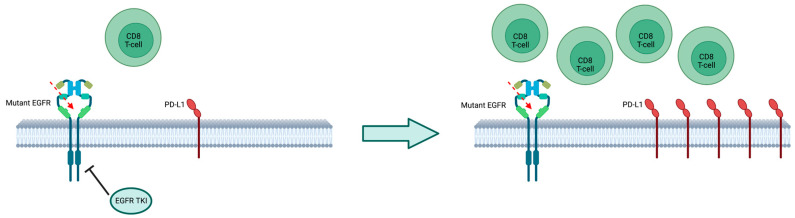
Tumor microenvironment after treatment with TKI; treatment with EGFR-TKI changes the tumor microenvironment by increasing PD-L1 expression. PD-L1: Programmed death-ligand-1.

**Figure 4 ijms-24-08878-f004:**
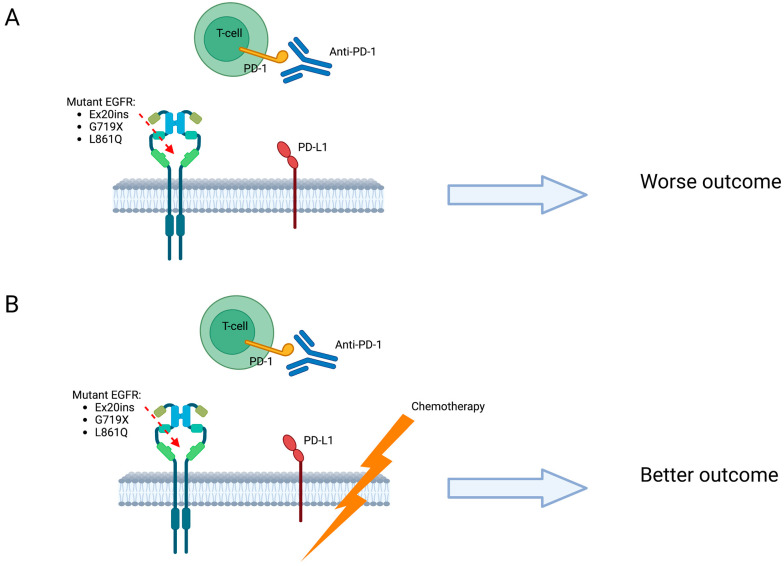
Immunotherapy VS. chemoimmunotherapy: Patients treated with ICIs (**A**), compared to chemotherapy plus ICIs (**B**), had a shorter PFS and OS, but the differences were not statistically significant. PD-L1: Programmed death-ligand 1. PD-1: Programmed death-1.

**Table 1 ijms-24-08878-t001:** Compound EGFR mutations and treatment [42].

Compound Mutation	Response to Treatment	Proposed Treatment
Combined common EGFR mutations (ex21 p.L858R + ex19del)	RR ≥ 75% with either first or second-generation TKIs	First or second-generation TKIs
Combined common (ex21 p.L858R + ex19del) plus uncommon EGFR mutations (any but ex21 p.L858R, ex19del or de novo ex20 p.T790M)	RR 40–80% and 100% with first-generation TKIs and afatinib	Afatinib
Combined uncommon EGFR mutations	RR 20–70%, ~80% and ~75% with first-generation TKIs, afatinib and osimertinib, respectively	Afatinib
Combined EGFR mutation (any) plus de novo ex20 p.T790M	Primary resistance to first- and second-generation EGFR-TKIs; osimertinib (RR 33.3%, DCR 100%)	Osimertinib

Ex: exon; Del: deletion; RR: Response rate.

**Table 2 ijms-24-08878-t002:** Ex20 ins inhibitors.

Ex20 Inhibitor	Trial	Toxicity	Response to Treatment
Poziotinib	NCT03066206	Diarrhea 92%, skin rash 90%, oral mucositis 68%, paronychia 68%, dry skin 60% (66% of G3 AEs on EAP)	ORR 32%, mPFS 5.5 m, mOS 19.2 m ORR of 46% and 0% in near (aa A767 to P772) vs. far loop ins
Mobocertinib	NCT02716116	PPP cohort 69% G ≥ 3 AEs 46% SAE EXCLAIM cohort 66% G ≥ 3 AEs 44% SAE	PPP cohort ORR 28% by IRC and 35% by investigator assessment, mPFS 7.3 m by IRC, mOS 24.0 m EXCLAIM cohort ORR 25% by IRC and 32% by investigator assessment
Amivantamab	NCT02609776	At RP2D 39% G ≥ 3 AEs, 31% SAE	ORR 40%, mPFS 8.3 m
Sunvozertinib (DZD9008)	NCT03974022 and CTR20192097	Most common TEAEs: diarrhea (G3 5.2%) and skin rash (G3 1%)	ORR 39.3% across all dose levels; dose level of 300 mg once daily, ORR 48.4% and DCR 90.3%
CLN-081 (TAS6417)	NCT04036682	Constipation 8%, diarrhea 8%, dizziness 8%, fatigue 8%, and chest pain 8%	5 evaluable pts: 2 pts PR, 3 pts SD
Tarloxotinib	NCT03805841	G3 TEAEs: prolonged QTc 34.8%, rash 4.3%, diarrhea 4.3%, increased ALT 4.3%	DCR 60%

AEs: adverse events; EAP: expanded access program; ORR: objective response rate; mPFS: median progression-free survival; mOS: median overall survival; aa: amino acids; PPP: platinum-pretreated patients; SAE: severe adverse event; IRC: independent review committee; RP2D: recommended phase II dose; PR: partial response; SD: stable disease; TEAEs: treatment-emergent adverse events; DCR: disease control rate.

## Data Availability

Not applicable.

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
