# Peer review of "Overview on Therapeutic Options in Uncommon EGFR Mutant Non-Small Cell Lung Cancer (NSCLC): New Lights for an Unmet Medical Need"

_ijms, 2023, doi:10.3390/ijms24108878_

Round 1

Reviewer 1 Report

Overall well structured and maintaining a flow of information in the following sections.

There are some specific point of concern as specified below:

1. In the introduction, the author may provide demographic data on lung cancer. Comparison of a developed country with a developing country, any observed pattern among incidences, death, age, sex.

2. Author may use EGFR full form.

3. In line no. 47 "Ranging between...... Different series" there is some typography error in the digit.

4. Line no. 88-96 from the same citation, however same citation is used in consecutive lines.

5. In line no. 106, "...... increasing during the last year. The author may rephrase the statement or may use the "last decade" or "no. of years". As NGS uses in mutation diagnosis or study for many years.

6. Line no. 122 typography error "Pirosequencing".

7. Line no 143, EMSO stands for......

8. Line no 222-227, 234-237, 246-250, 385-390, 392-394, 408-411, 440-443, 447-450, 499-502, and 505-508: Same citation used in consecutive lines.

9. Line no 253 and 255 from the same article, Author may amend a single line with the next paragraph. 

10. In conclusion section "In the next future....". The statement can be rephrased.

Very steady flow of information throughout the article.

Minor changes in few sentences are required, which I observed.

Reviewer 2 Report

Very well written article by the authors on treatment options for uncommon EGFR mutant NSCLC.  Just some minor revisions:

Line 30-31 could be rewritten clearly

Acronyms like ESMO, OS, ORR etc to be expanded when used the first time

Line 435 needs to be checked

Figure 2: I am not sure whether the arrow connecting the 2 figures is needed

Fig 4: The authors says that "The group of patients treated with ICIs, compared to chemotherapy plus ICIs, 523 had a longer PFS although there was no significant difference in OS" but in the figure, it says worse outcome for ICI only, which is counter intuitive

Minor grammatical errors and/or spell checks required

Reviewer 3 Report

Pretelli et al. described existing data on outcomes, epidemiology, and clinical characteristics of lung cancer patients with rare EGFR mutations, with a focus on intracranial activity and response to immunotherapy. The topic is very interesting, but there are several concerns that should be addressed.

  1. The reference style should be carefully revised throughout the entire manuscript.
  2. The structure of the review needs to be rearranged, as I noted that the authors sometimes considered every three lines a separate paragraph, which makes no sense.
  3. Line 35: "The major achievements in the field of EGFR tyrosine kinase inhibitors 35 (TKIs) have been obtained in patients with NSCLC", reference should be added.
  4. Line 79 a p respectively should be corrected.
  5. The abbreviations should be revised carefully.
  6. There are many grammar mistakes; the review should be revised carefully.
